# communications
# engineering

# Degradation in parallel-connected lithium-ion battery packs under thermal gradients

Max Naylor Marlow[1], Jingyi Chen[1] & Billy Wu [1✉]

Practical lithium-ion battery systems require parallelisation of tens to hundreds of cells, however understanding of how pack-level thermal gradients influence lifetime performance remains a research gap. Here we present an experimental study of surface cooled parallel-string battery packs (temperature range 20–45 °C), and identify two main operational modes; convergent degradation with homogeneous temperatures, and (the more detrimental) divergent degradation driven by thermal gradients. We attribute the divergent case to the, often overlooked, cathode impedance growth. This was negatively correlated with temperature and can cause positive feedback where the impedance of cells in parallel diverge over time; increasing heterogeneous current and state-of-charge distributions. These conclusions are supported by current distribution measurements, decoupled impedance measurements and degradation mode analysis. From this, mechanistic explanations are proposed, alongside a publicly available aging dataset, which highlights the critical role of capturing cathode degradation in parallel-connected batteries; a key insight for battery pack developers.

[1] Dyson School of Design Engineering, Imperial College London, London SW7 2AZ, UK. ✉email: billy.wu@imperial.ac.uk

Driven by the accelerating uptake of electric vehicles, a dramatic increase in the usage of lithium-ion batteries (LIB) has occured. However, individual LIBs have low voltages and relatively small capacities; driving the need to connect cells in series and parallel to create high voltage, large capacity battery packs. Whilst it is usually assumed that parallel connected cells will experience identical operating currents during operation, in practice a range of studies have identified significant heterogeneities in operating current and temperature during cycling[1–3]. These effects are caused by variability between cells, thermal gradients, and uneven potential drops caused by interconnection resistances causing heterogeneous current distributions and state-of-charge (SOC) imbalances between cells. In recent years, researchers have published a range of studies detailing the effects of operating conditions[4], cell-to-cell capacity and impedance variations[5–8], topology[9,10] and thermal gradients[6,10–13] on parallel string current distributions and the instantaneous performance of cells and packs. However, there is limited work addressing the lifetime implications of such heterogeneous conditions, and thus a lack of underpinning understanding of the interplay between pack design factors and degradation modes; motivating this work.

Considering the implications of heterogeneities on pack degradation, experimental investigation of 1S2P packs (1 in series, 2 in parallel) with deliberately mismatched cell impedance has been shown to lead to a maximum reduction in lifetime of approximately 40% when comparing balanced and imbalanced (20% impedance difference) packs[14], attributed to the uneven peak currents applied to each cell due to the deliberate impedance mismatch. Investigation of 1S2P configuration large format 60 Ah prismatic lithium iron phosphate cells, under applied thermal gradients has shown that the application of a 30 °C thermal gradient (temperature range 25 °C to 50 °C) applied leads to a doubling of the degradation rate[15]. Over 1000 cycles, the initial deviation in currents was higher for the heterogeneous temperature pack, and continued to diverge, while current divergence was reduced in the homogeneous temperature pack. This led to a 50% reduction in capacity loss at 1000 cycles in the heterogeneous case with applied thermal gradient. Interestingly, the high temperature cell within the parallel string degraded slower than a reference cell cycled at the same temperature outside of a parallel string. This differential aging in both cases was attributed to the imbalanced currents present in the parallel strings, in particular the high peaks identified at start and end of discharge.

Cycling cells with deliberately induced start-of-life (SOL) deviations in cell capacity and impedance, but in the absence of applied thermal gradients, shows that total cell capacity fade and impedance growth converged due to the different cell currents and therefore variable total current throughput, with cells experiencing a higher current undergoing accelerated degradation[7]. Investigation of 1S2P coupled cells harvested from an aged electric vehicle battery pack found that after aging in-service, significant increases to parameter spread occurred, with a mean increase in capacity spread of 0.5%, and 3.5% in internal resistance, with several cell pair resistance spreads increasing to over 10%[16]. Further cycling of selected modules indicated that rather than converging due to self-balancing over time, the heterogeneities in cell performance increased. Increasing parallelisation reduces the degree of heterogeneity experienced by strings, particularly reducing peak currents at the end of discharge[17]. This was suggested to be a factor prolonging lifetime of highly parallel packs, reducing the impact of either low capacity or high impedance cells.

Model-based investigation of thermal gradients applied to 1S6P parallel strings, simulating degradation via solid-electrolyte interphase (SEI) layer growth leading to capacity loss and power fade[6], indicates that the degradation rate of the string is linked to the applied thermal gradient, predicting a 10% increase in rate of capacity fade when comparing ±25 °C thermal gradients. This is shown experimentally by applying a 20 °C thermal gradient to 1S2P parallel cell strings, which leads to increasing cell-to-cell capacity deficits[18], and shows that if thermal boundary conditions are similar, convergence will occur when the cells exhibit either decelerating or linear degradation curves, whilst divergence is expected with an accelerating degradation curve. Other researchers have compared the degradation rate to a single cell to a 1S5P pack configured as a cell stack with the outer surfaces of cells 1 and 5 convectively cooled[19]. This lead to highly non-uniform degradation within the string, which degraded significantly faster than the baseline cell (7% greater capacity loss at 2215 cycles). This is attributed to the higher mean cycling temperatures of the pack cells compared to the baseline single cell, with a peak temperature rise 12.5 °C higher in the parallel string.

Divergent behaviour is undesirable as it may lead to extreme currents in areas of packs, leading to reduced lifetime, extreme peak currents and safety concerns[2,10]. Some works have indicated that when thermal conditions are consistent and degradation follows the commonly seen linear or convex pathway, initially diverging cells will converge to identical capacities[20], whilst other works suggest that thermal gradients within packs will lead to divergent cell performance over time[6,15,18]. These differing conclusion motivate this work, whereby an experimental approach is used to investigate lifetime performance within parallel-connected LIBs, and characterisation of individual cells within cycled packs enables application of detailed diagnostics to identify degradation modes occurring within individual cells. Key insights include identification of cathode impedance growth as a key contributor to the increasingly heterogeneous current distributions observed, which is rarely discussed in the academic literature. Meanwhile, thermal gradients are shown to lead to increasingly heterogeneous behaviour in operating regimes where power fade rates are negatively correlated with cell temperature. These insights thus inform the proposed mechanistic model, which provides key insights for battery pack developers.

## Materials and methods

**Cell selection and grouping**. The cells selected in this study were pouch cells manufactured by Dow Kokam, with a nominal capacity of 5 Ah; the SLPB11543140H5 cell. These are high-power cells containing a blended lithium nickel manganese cobalt oxide (NMC) and lithium cobalt oxide (LCO) cathode and a graphite anode. The nominal characteristics, of the SLPB11543140H5 (hereby referred to as the 'Kokam 5 Ah') are detailed in Table S1.

As previously identified[5,6,16,21,22], small variations in cell capacity and impedance may lead to variations in lifetime performance. To reduce any influence of cell-to-cell variations, differences in capacity and resistance between the cells, within each pack, were minimised. 19 cells from the same batch, stored under identical conditions, were selected and initially characterised. SOL discharge capacity was measured at 0.04 C and impedance was measured using Electrochemical Impedance Spectroscopy (EIS). To obtain the SOL discharge capacity, the cells were initially charged using a CC-CV charging protocol (1 C charge to 4.2 V, current cutoff of 0.01 C), rested for 60 mins and subsequent discharged at 0.04 C to 2.7 V. The cells were then charged to 50% SOC, indicated by the prior discharge capacity measurement, and allowed to equilibrate for a minimum of 12 h. Subsequently, a single EIS measurement (taken using an Autolab PGSTAT201) was taken at 50% SOC with an RMS excitation current of 0.2 A (frequency range 2500–0.01 Hz, 10 frequencies

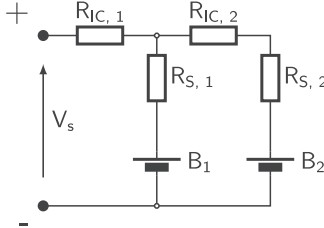

**Fig. 1 Circuit diagram of the 1S2P packs used during aging experiments.**
Here, $R_{IC}$, $R_S$, B and V denote the interconnection resistance, contact resistance, battery and the voltage of the parallel string, respectively.

logarithmically spaced per decade). In all cases, cells were characterised in a thermal chamber (Binder KB53) at 20.0 °C. The full batch SOL discharge capacity and impedance measurements are shown in Supplementary Fig. S1. To group the cells, the capacity and series resistance ($R_0$) were evaluated. The highest $R_0$ cells were discarded and the remaining twelve cells were used produce six 1S2P packs. These cells were sequentially grouped by their corresponding $R_0$ values to leave six pairs (Supplementary Fig. S1e–j). All $R_0$ differences were minimised to below 0.45% (5.72 $\mu\Omega$) and capacity differences to below 1.314% (65.6 mAh).

**Test bench development and pack topology**. To investigate the effects of thermal gradients on parallel string lifetime, a test bench was developed for cycling 6 1S2P packs under controlled thermal boundary conditions. Recent studies have used Peltier elements to control LIB thermal boundary conditions with a high degree of control[11,13,23,24]. The test bench was designed to allow for simple disassembly and individual cell characterisation during cycling experiments. Further details of the test bench design and calibration are found in Supplementary Methods Section 1.2. A 1S2P parallel U-configuration pack was selected as a test case. The electrical topology, shown in Fig. 1, used equal interconnection and current sense resistances of 1 mΩ. To ensure consistency, all busbars and current sense resistors were calibrated using a Bio-logic VMP-3 potentiostat. The resistors used (Bourns CSS4J-4026R-1L00F and CSS4J-4026R-2L00F) were selected due to their low temperature coefficient of 75 ppm/K, minimising the impact of resistive heating on current measurement accuracy. Once calibrated, the voltage drop across each current sense resistor was recorded using a datalogger (PicoLog TC-08).

Cell testing was carried out using a quick-release test fixture (Supplementary Fig. S6). All cell-level intermediate characterisation was carried out in a thermal chamber (Binder KB53) at 20.0 °C. In all cases, contacts were cleaned before connections were made using an abrasive (Medium Grit 3M Scotchbrite Abrasive Pads) and cleaning with acetone and isopropanol. Cell tabs were cleaned using acetone and isopropanol. After assembly of either pack or single cell test fixtures, the contact resistance was measured using a Keithley 2182A Bench Digital Multimeter, ensuring consistency of contact resistances throughout testing.

**Cycling and characterisation procedure**. The thermal gradients selected for investigation were 0, −12.5 and ± 25 °C. These values were selected based on thermal gradients identified in other experimental works[4,10,19]. The selected thermal cases are summarised in Table 1. Overall, 3 packs were cycled at a homogeneous surface temperature of 20.0, 32.5 and 45.0 °C, while 3 packs were cycled under thermal gradients with a constant mean temperature of 32.5 °C and applied thermal gradients of +25 °C, −12.5 °C and −25 °C. Here, positive/negative thermal gradients are defined with respect to the pack terminals; a positive gradient

means the cell further from the terminals (B2 in Fig. 1) is at higher temperature, and vice versa for a negative gradient.

After grouping, each cell was individually characterised as follows. All cell-level characterisation tests were carried out using a BaSyTec 60 A XCTS G2 battery cycler and cells were allowed to thermally equilibrate for 180 mins at 20 °C before testing. For all cells, the capacity was measured at 0.1, 1 and 2 C, referenced to the nominal cell capacity of 5 Ah. Typically, the cell was charged (1 C CC-CV charge to 4.2 V, current cutoff of 0.01 C), rested for 60 mins before discharging to 2.7 V at the selected rate. Hybrid pulse power characterisation (HPPC) measurements were taken using a modified HPPC protocol[25]. In the protocol used, initially each cell was charged (1 C CC-CV charge to 4.2 V, current cutoff of 0.01 C), before discharging at 0.25 C to 90% SOC (referenced against the nominal cell capacity). Each cell was then allowed to equilibrate for 60 min before 18 s duration charge and discharge pulses of 0.2, 1 and 3 C were applied, with a 60 second relaxation period between pulses. Subsequently, the cells were discharged at 0.25 C for another 10% SOC to the next breakpoint, and this process was repeated until either the cell voltage reached 2.7 V or 9 sets of pulses were applied. Subsequently, the packs were assembled and the capacity characterisation was repeated for the as-assembled pack, held at a constant surface temperature of 20 °C using the pack test bench. All pack-level experiments were carried out using a BaSyTec 50 A XCTS G1 battery cycler. After SOL characterisation, the packs were cycled using a constant current 2 C cycling regime, referenced to the nominal pack capacity of 10 Ah. During each run, the packs were initially charged (1 C CC-CV charge to 4.2 V, current cutoff of 0.01 C), before being allowed to rest for 60 mins before start of cycling. After 125 cycles at 2 C, each pack was disassembled and the SOL capacity and HPPC characterisation procedure was repeated for each individual cell, before reassembly of the pack and repetition of the SOL pack capacity characterisation.

These cycling and characterisation processes were then repeated for 16 subsequent breakpoints until each pack had experienced 2000 total 2 C cycles. Here, due to the COVID-19 pandemic, a break of 3 months occurred between the initial 125 cycles and subsequent experiments. Cells were stored at <10 °C for this period and no identifiable impacts on overall cycling performance were observed. At end-of-life (EOL), a repeat set of EIS measurements were taken to verify observed impedance changes, using an identical procedure to that used to initially group cells, taken at the same cell OCV (±2 mV).

**Data analysis**. Data analysis was carried out using MATLAB 2020a, which was used to synchronise and analyse all data acquired. During the degradation experiments, several intermittent power failures in the laboratory led to some brief (<24 h) irretrievable data losses from the pack current dataloggers, which is naturally excluded from the analysis, including incomplete cycle data. For the heatmaps presented in Section "Current Distribution Evolution", the gaps were filled with constant values from the last recorded full cycle, and has a negligible impact as the lost data accounted for <2% of the dataset (Supplementary Table S2 identifies the missing data).

For Incremental Capacity Analysis, the discharge data at 0.1 C was selected to provide the highest quality data and was analysed using the LEAN algorithm[26]. Here, the smoothing filter used was $\alpha = [0.1059, 0.121, 0.1745, 0.1972, 0.1745, 0.121, 0.1059]^T$, whilst the filtering matrix, $A_m$ was unmodified[26]. Parameterisation of the diagnostic fitting model followed the procedures laid out by refs. [27,28] and is summarised in Supplementary Section 2.3 along with the model structure. All processing and fitting of EIS data was carried out using the ZView software package from Scribner Associates.

**Table 1 Pack aging thermal conditions and identifiers.**

| Pair No. | Pack Name | B1 Temperature [°C] | B2 Temperature [°C] | Thermal Gradient [°C] | Mean Temperature [°C] |
|---|---|---|---|---|---|
| 1 | P20.0/0.0 | 20.0 | 20.0 | 0.0 | 20.0 |
| 2 | P45.0/0.0 | 45.0 | 45.0 | 0.0 | 45.0 |
| 3 | P32.5/+25.0 | 20.0 | 45.0 | +25.0 | 32.5 |
| 4 | P32.5/0 | 32.5 | 32.5 | 0.0 | 32.5 |
| 5 | P32.5/-25.0 | 45.0 | 20.0 | -25.0 | 32.5 |
| 6 | P32.5/-12.5 | 38.8 | 26.3 | -12.5 | 32.5 |

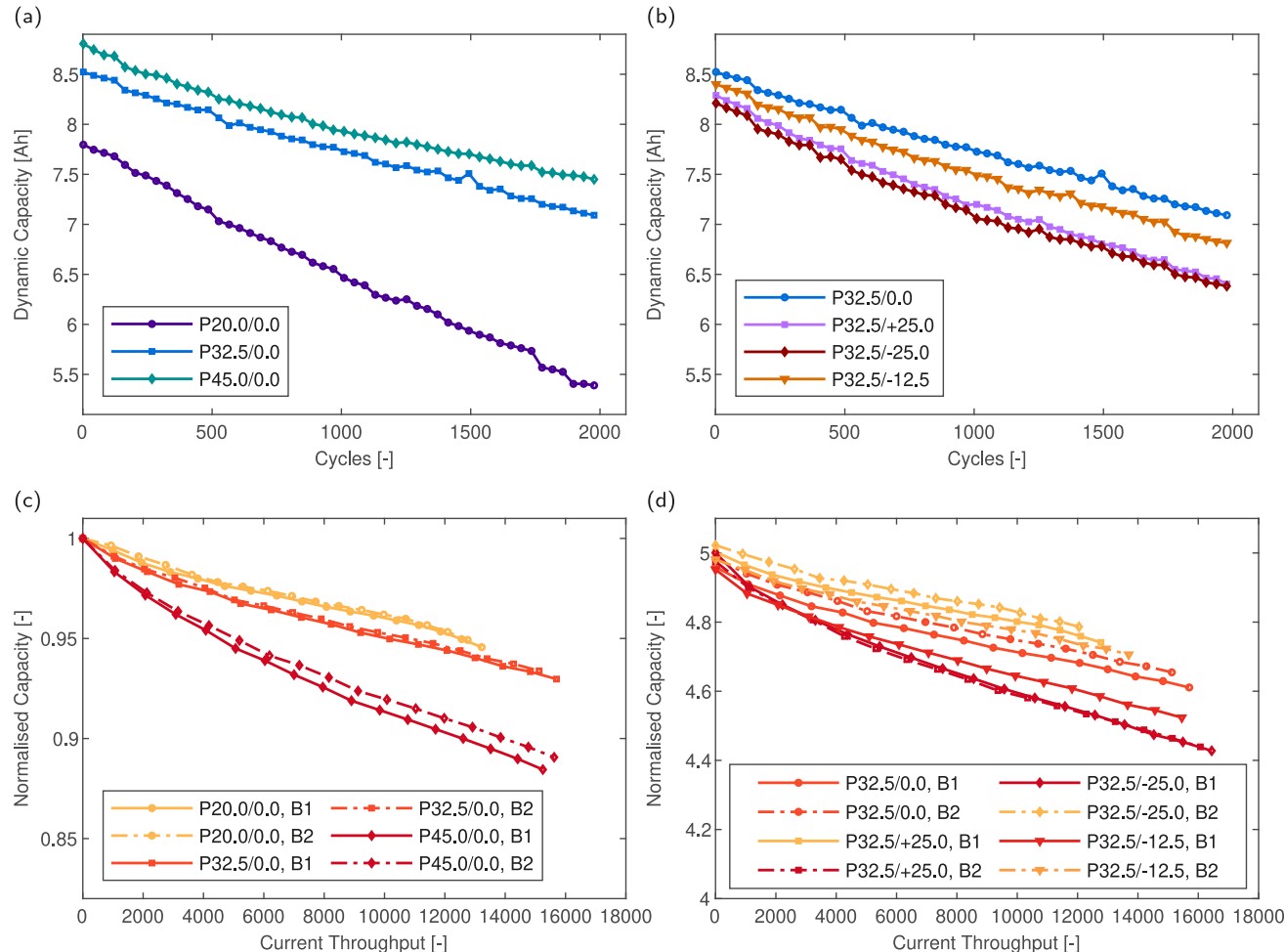

**Fig. 2 Capacity fade of cells and packs under different thermal conditions.** Dynamic discharge capacity during aging versus cycle count for homogeneous temperature (**a**) and heterogeneous (**b**) temperature packs. Normalised static measured cell capacity (at 0.1 C, 20 °C) versus current throughput for all homogeneous temperature (**c**) packs and heterogeneous temperature (**d**) packs. Note that in (**c**) and (**d**), trace colour is proportional to cell cycling temperature. Absolute static capacity values for (**c**) and (**d**) can be found in Supplementary Fig. S7.

## Results and discussion

**Capacity fade and resistance growth**. After cycling (2000 cycles/pack) the pack capacity loss was evaluated. Figure 2a, b shows the rate of dynamic capacity loss in the homogeneous and heterogeneous temperature cases respectively, plotted every 20 cycles (N.B. data from P32.5/0.0 is repeated in Fig. 2b for ease of comparison). The rate of dynamic capacity loss reflects the accessible pack capacity under continuous 2 C cycling rather than the underlying cell capacity. The dynamic or accessible capacity from a parallel string is a function of several factors including the underlying cell capacity, cell impedance and the cell open circuit voltage (OCV). At decreased cell temperatures, increasing cell impedance is the cause of branch resistance change and, therefore, current heterogeneities during a single cycle. Greater current

heterogeneities during a single cycle imply a wider cell-to-cell SOC distribution and consequently decreased accessible string capacity. However, the interplay between these factors becomes more complex when considering performance over the lifetime of the string. There is also a significant deficit between the accessible capacity during dynamic cycling and the static reference capacity measured, as shown in Fig. 2c, d for the homogeneous and heterogeneous temperature packs, respectively.

For example, after 2000 cycles, P20.0/0.0 loses approximately 25% accessible capacity, however the static capacity loss is consistent between cells and low (~5% loss). In comparison, P45.0/0.0 shows the smallest reduction in dynamic accessible capacity (~11%), which is close to the reference capacity loss of ~13%. This is a consequence of the reduced cell impedance

during cycling at elevated temperatures allowing a greater portion of cell static capacity to be accessed per cycle. Meanwhile, for P32.5/+25.0 and P32.5/−25.0 an accelerated loss in overall accessible capacity is observed when compared to P32.5/0.0 (Fig. 2b). Decreasing the magnitude of the thermal gradient reduces the loss of accessible capacity by 65% when comparing P32.5/−25.0 to P32.5/−12.5, implying that increasing magnitudes of thermal gradient are increasingly detrimental to both overall performance and pack lifetime. However, minimal difference is seen when comparing the positive and negative thermal gradient cases. Considering the reference capacity loss in the heterogeneous temperature packs (Fig. 2d), significant deviations in static capacity loss occurred, proportional to the cell temperature. No significant deviation in the rate of static capacity loss is seen when comparing the direction of the applied thermal gradient. Notably minimal differences in capacity loss rate occur in cells cycled at <32.5 °C, while the rate of degradation accelerates significantly above 32.5 °C.

During cycling, cell HPPC resistance was measured to track DC resistance growth. Figure 3a–f presents the extracted HPPC resistance values $R_{DC|0.05s}$ and $R_{DC|5.0s}$ for all cells at 50% SOC. The magnitude of $R_{DC|0.05s}$ is approximately equal at SOL for all cells and is used to approximate $R_0$ and $R_{SEI}$. While the magnitude of $R_{DC|0.05s}$ is approximately 25% above that measured using EIS, it serves as a useful approximation to track the lumped short time constant resistances (an approach which has been validated[29]). Meanwhile at SOL, $R_{DC|5.0s}$ is consistent between cells and accurately approximates the sum of $R_0$, $R_{SEI}$ and $R_{CT}$. SOL and EOL EIS measurements were used to select the evaluation time for each approximation and verified that the selected times were valid approximations of the cell impedance magnitude in both cases.

As shown in Fig. 3a–f, the growth of $R_{DC|0.05s}$ is relatively insensitive to the pack or cell considered, with a linear growth in all cases of between 35 and 48% over 2000 cycles. However, divergence is seen in the longer time constant DC resistance, $R_{DC|5.0s}$. The rate of growth in $R_{DC|5.0s}$ is greater at lower temperatures, for example, in P20.0/0.0 (Fig. 3a). In comparison for P45.0/0.0 the rate of growth in $R_{DC|5.0s}$ slows (Fig. 3e). The intermediate P32.5/0.0 shows linear growth in $R_{DC|5.0s}$. Typically, it is assumed that the rate of impedance growth will increases at higher temperatures, which would occur if SEI layer growth was the dominant power fade mechanism. However, Fig. 3a–f indicates the rate of $R_{CT}$ growth is inversely proportional to temperature. Further confirmation of these observations is shown in the packs cycled under thermal gradients. P32.5/+25.0 shows divergent growth in $R_{DC|5.0s}$, with significant divergence between B1 (at 20.0 °C) and B2 (at 45 °C) above 7.5 kAh current throughput (Fig. 3b). This leads to an eventual deviation in $R_{DC|5.0s}$ of 20%. Similar behaviour is observed in P32.5/−12.5 and P32.5/−25.0 (Fig. 3d, f).

In order to provide confirmation of these observations, EIS measurements were taken at the SOL and EOL for each cell at 50% SOC with respect to measured cell capacity at SOL and EOL. The measured spectra are shown in Fig. 4a–f. The trends highlighted by the HPPC measurements are confirmed with the EIS measurements. First considering P20.0/0.0, significant divergence at EOL is seen in the impedance magnitude below approximately 20 Hz, while the high frequency impedance response remains relatively constant, with negligible divergence seen between the cell pairs. Considering the homogeneous temperature cases (Fig. 4a, c, e) it can be seen that the magnitude of the $R_{CT}$-associated semicircle is inversely proportional to the applied temperature. Divergence between the spectra is seen in P20.0/0.0 and P32.5/0.0, however, in P45.0/0.0 impedance is not seen to diverge significantly. The homogeneous temperature cases

indicate $R_0$ may grow slightly at elevated temperatures. Impedance responses for P32.5/+25.0, P32.5/−12.5 and P32.5/+25.0 are shown in Fig. 4b, d, f, and follow similar trends. Again, the lower temperature cells experience larger increases in $R_{CT}$, while the high temperature cells see greater growth in high frequency impedances. These EIS measurements also serve to demonstrate the validity of approximating total cell impedance using DC resistance measurements taken with different time intervals.

Distribution of Relaxation Times analysis indicates that in NMC cells, the total charge transfer resistance is dominated by the cathode charge transfer resistance, which also increased significantly faster than the anode charge transfer resistance[30]. This increase was found to be inversely proportional to temperature and it was proposed that the rise in cathode charge transfer resistance is caused by particle fracture and consequent loss of electrochemically active material. Meanwhile EIS analysis of aged cells with inserted Lithium-metal reference electrodes shows that the impedance rise during cycling is predominantly caused by changes to the cathode-electrolyte interphase resistance and cathode charge transfer resistance[31]. In comparison, changes in anode impedance lead to minimal effects on the full cell; with growth in cathode impedance the dominant change at the full cell level. Consequently, the impedance growth demonstrated in cycled cells is likely dominated by cathode degradation leading to increases in cathode charge transfer resistance, and this is indicative of cathode capacity loss due to loss of electrochemically active sites in the cathode.

**Current distribution evolution.** In this section, the influence of capacity loss and impedance changes on the heterogeneous behaviour within the pack is analysed, with the aim of attributing specific changes in overall behaviour to cell-level changes within the pack, and subsequently identifying underpinning mechanisms that may be generalised to other systems. Figure 5 visualises the measured current distribution in the homogeneous temperature packs studied, plotted against cycles and depth of discharge (depth of discharge is defined as the current passed, divided by the total current throughput per cycle). Minor variations every 125 cycles are caused by unavoidable slight variations in contact resistance during assembly and disassembly of the packs. The mechanisms driving uneven current distributions at SOL have been investigated in other works[4–6,10,15] and is not the focus of this work, which is instead on the evolution of current distributions during aging.

In the homogeneous temperature cases, the dominant and accelerated growth of $R_{CT}$ was identified and leads to convergence of pack currents. As the packs are cycled at constant temperature, the cell impedance is approximately equal at SOL, meaning the initial current distribution is dictated by the interconnection resistance. Subsequently during aging, accelerated growth in $R_{CT}$ at lower temperatures leads to convergence, as total branch resistances become more balanced and the impacts of the interconnection resistance is reduced by the increased magnitude of the cell impedance, which decreases the ratio of cell to interconnection resistance. Using P20.0/0.0 as an example (Fig. 5a–c), initial current peaks in B1 occur below 0.1 depth-of-discharge (DoD), peaking at over 2.2 C. Minimal rebalancing of currents occurs, attributed to the increased impedance at low temperatures increasing cell overpotentials and therefore reducing the opportunity for rebalancing at higher DoDs. During aging, the current distribution is homogenised, with the initial high peak current region steadily decreasing in duration. The peak current diffence over life is reduced from approximately ± 2 A at SOL to < ± 1 A at EOL

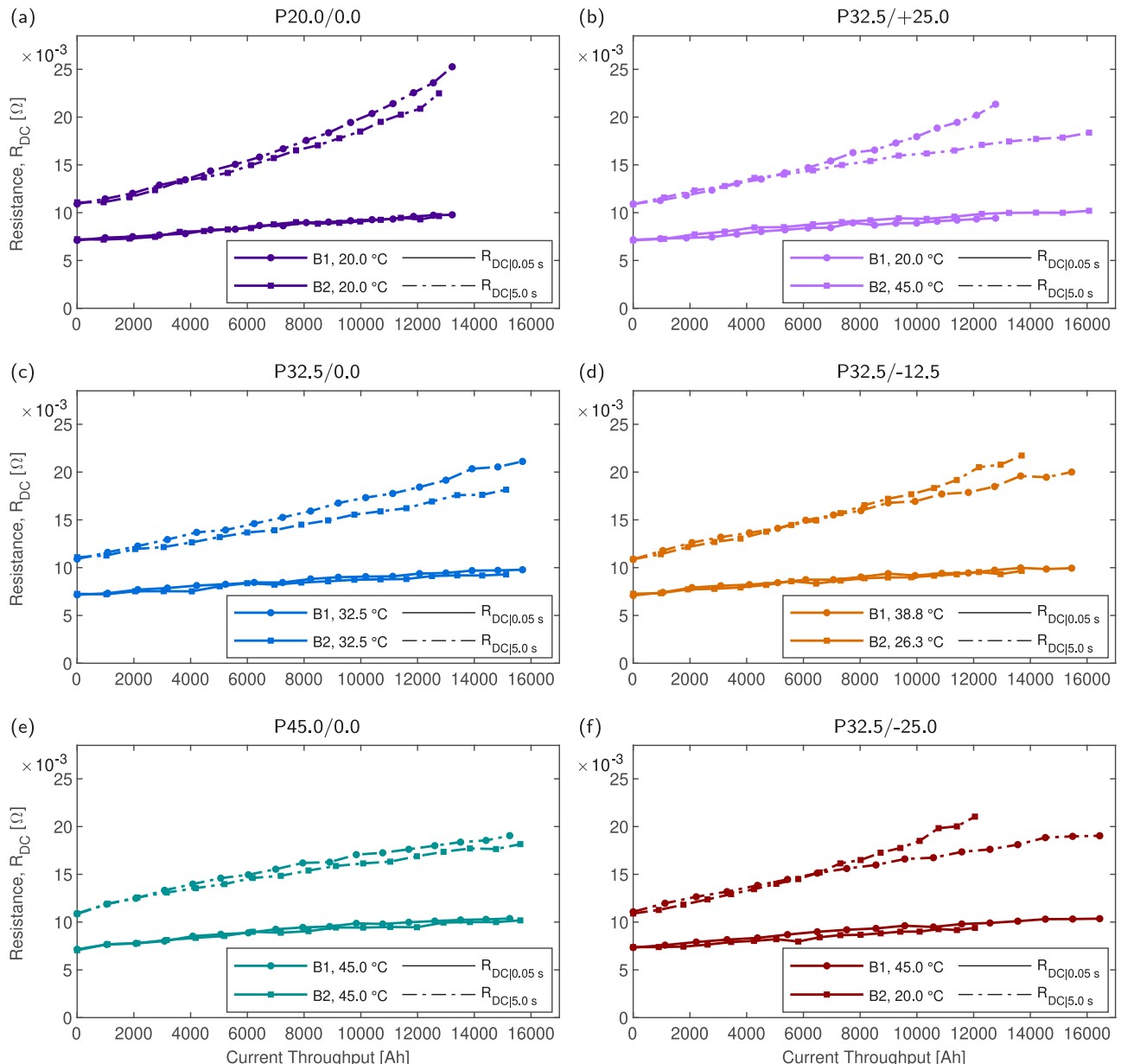

**Fig. 3 Growth in measured hybrid pulse power characterisation (HPPC) resistances, $R_{DC|0.05s}$ and $R_{DC|5.0s}$, for all packs.** All HPPC measurements taken at 20 °C, 1 C pulse discharge. **a** P20.0/0.0. **b** P32.5/+25.0. **c** P32.5/0.0. **d** P32.5/−12.5. **e** P45.0/0.0. **f** P32.5/−25.0.

In P45.0/0.0 (Fig. 5g–i) convergence occurs, although less significantly than at lower temperatures. Due to the non-linearly decreasing impedance at elevated temperature; interconnect resistances have a greater impact on the current distribution, reducing convergence. The convergence observed is principally in the initial region below 0.1 DoD, and this is attributed to the increasing $R_{SEI}$ in the high temperature pack during aging. Increases in $R_{SEI}$ occur above 40 °C (Supplementary Fig. S15) and led to decreasing initial peak currents and therefore initial SOC deviation, as the high frequency branch impedance is increased at start of discharge. More significant rebalancing occurs within the high temperature pack at low DoD due to the decreased cell overpotentials. As the overall current distribution is only partially convergent over time, this rebalancing remains at EOL.

Figure 6 visualises the measured current distribution in the heterogeneous temperature packs. In P32.5/+25.0 (Fig. 6a–c), the current is initially higher in B2, as the thermal gradient direction reduces impedance of B2, leading to a lower overall branch impedance in branch 2. Here, the overall current distribution is divergent during aging, as the previously identified accelerated impedance growth in B1 at low temperature combined with the lower apparent cell impedance of B2 results in an increasing mismatch in branch resistances. Due to the changes in cell impedance, the resistance of branch 1 increased more rapidly during cycling. As shown in Fig. 3, the measured cell DC resistance diverged at approximately 1000 cycles, and this increase in mismatch of branch resistances is observed in the increasing magnitude of the current distribution above 1000 cycles.

In the converse case, P32.5/−25.0 (Fig. 6g–i), over the life of the pack the current distribution is divergent, again driven by the accelerated impedance growth seen in the low temperature cell. Whilst for some areas of the SOC window the current difference decreases during cycling (for example, between 0.6 and 0.8 DoD in P32.5/−25.0, shown in Fig. 6i), the mean difference in current increases in all cases with applied thermal gradients. The

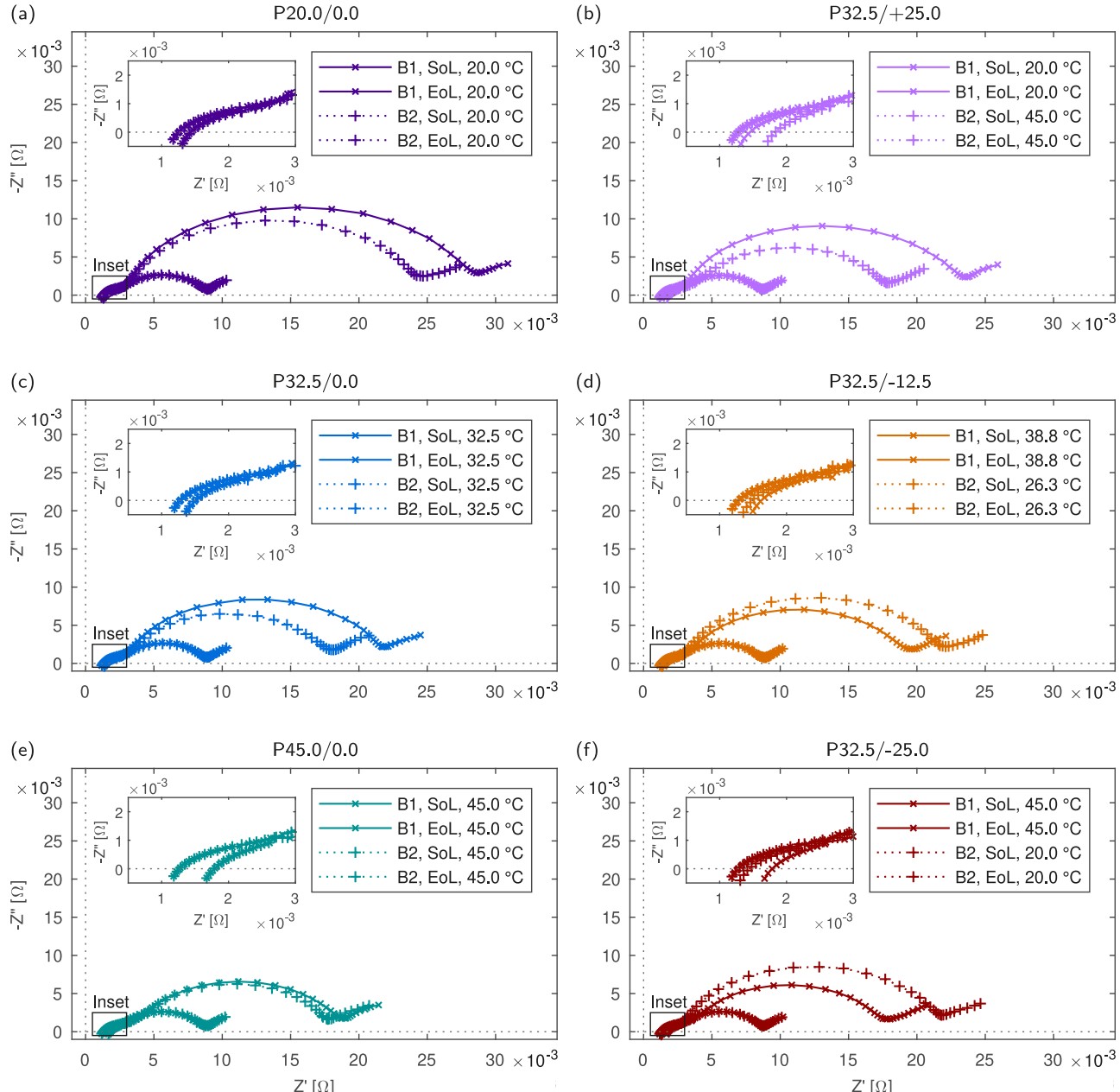

**Fig. 4 Electrochemical impedance spectroscopy (EIS) spectra at start-of-life (SOL) and end-of-life (EOL) for all packs.** Measurements were taken at 50% state-of-charge (SOC) (3.78 V ± 0.002 V), 20 °C from 10 kHz–0.01 Hz, 250 mA excitation current. **a** P20.0/0.0. **b** P32.5/+25.0. **c** P32.5/0.0. **d** P32.5/−12.5. **e** P45.0/0.0. **f** P32.5/-25.0.

divergence in cell impedance occurring at approximately 1000 cycles has a more significant impact on the current distribution, as interconnection resistances are no longer assisting in balancing branch resistances as seen in the positive thermal gradient case. In the moderate negative thermal gradient pack, P32.5/−12.5 (Fig. 6c, d), while currents are somewhat divergent during aging, this is significantly reduced compared to ±25.0 °C packs, attributed to the reduced difference in cell impedance induced by temperature, alongside the less significant divergence in cell impedances over the lifetime of the pack. In general, it can be concluded that larger thermal gradients will lead to increased heterogeneities over the pack lifetime, provided impedance growth is negatively correlated with cell temperature; leading to impedance divergence between cells. Conversely, if cell impedances are convergent and impedance growth is positively

correlated with temperature, heterogeneities in current will be reduced. This highlights the critical nature of understanding cell impedance growth behaviour over the expected operating temperatures during design of parallel cell strings.

Figure 7 shows a schematic representation of the effects of branch resistance growth on current distributions within a homogeneous temperature versus a heterogeneous temperature pack. Firstly in the homogeneous temperature case, at SOL (1) branch 2 has a higher total resistance and, therefore, a higher peak current and greater total current throughput passes through B1. By EOL (2), increases to the impedance of B1 leads to an overall equalisation of branch resistance and, therefore, homogenises the current and SOC distribution observed. Whilst the branch resistance is not constant throughout a single discharge, (3) illustrates the relationship between total branch resistance and

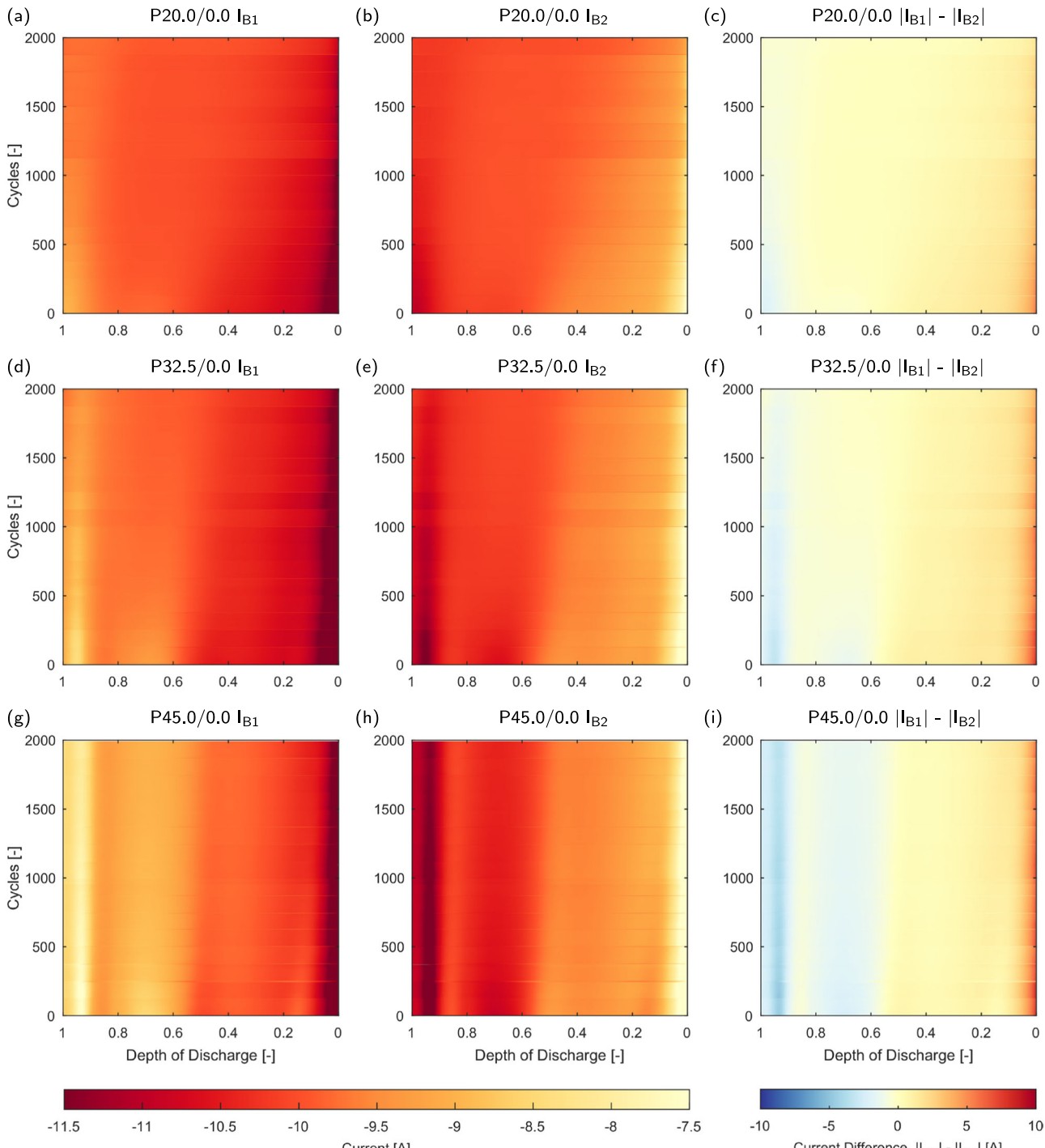

**Fig. 5 Heatmaps of current distribution within homogeneous temperature packs over lifetime.** Depth of discharge defined as current passed divided by the total current throughput per cycle. **a** P20.0/0.0 $I_{B1}$. **b** P20.0/0.0 $I_{B2}$. **c** P20.0/0.0 $|I_{B1}| - |I_{B2}|$. **d** P32.5/0.0 $I_{B1}$. **e** P32.5/0.0 $I_{B2}$. **f** P32.5/0.0 $|I_{B1}| - |I_{B2}|$. **g** P45.0/0.0 $I_{B1}$. **h** P45.0/0.0 $I_{B2}$. **i** P45.0/0.0 $|I_{B1}| - |I_{B2}|$.

mean current split. Schematically represented is the start of discharge total branch resistance and the differing rates of branch resistance increase leading to homogenisation of currents. Conversely in the heterogeneous temperature case, at SOL (4) the elevated temperature of B1 leads to a greater difference in total branch resistance, and therefore a greater split in cell currents. However, due to the previously observed accelerated cell impedance growth at low temperatures, this effect is compounded over the life of the pack, as the total resistance of branch 2 grows

faster than that of branch 1. Thus, by EOL (5), the mean current through branch 2 is reduced while branch 1 sees an increased mean current. A schematic diagram (6) illustrates the relationship between branch total resistance and mean current split.

Within parallel strings, accessible capacity is reduced by the EOD SOC deficit, i.e., the remaining unused cell capacity at EOD. The SOC defict occurs as one cell in the string is preferentially discharged when connected in parallel, leading to the pack reaching either the lower- or upper-cutoff voltage before other

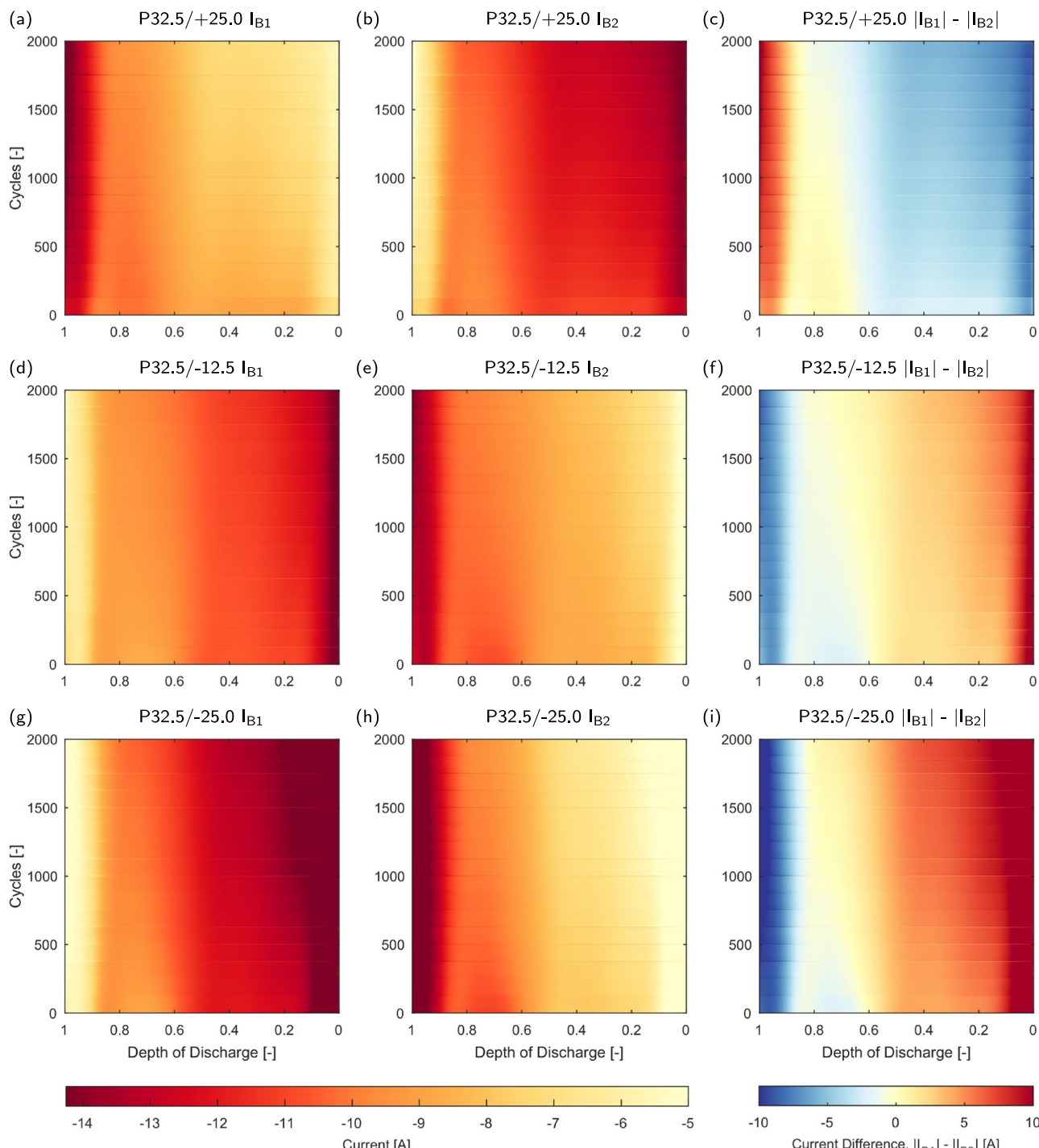

**Fig. 6 Heatmaps of current distribution within heterogeneous temperature packs over lifetime.** Depth of discharge defined as current passed divided by the total current throughput per cycle. **a** P32.5/+25.0 $I_{B1}$. **b** P32.5/+25.0 $I_{B2}$. **c** P32.5/+25.0 $|I_{B1}|$ - $|I_{B2}|$. **d** P32.5/−12.5 $I_{B1}$. **e** P32.5/−12.5 $I_{B2}$. **f** P32.5/−12.5 $|I_{B1}|$ - $|I_{B2}|$. **g** P32.5/-25.0 $I_{B1}$. **h** P32.5/-25.0 $I_{B2}$. **i** P32.5/−25.0 $|I_{B1}|$ - $|I_{B2}|$. Note colour scale in individual cell current heatmaps is different to that in Fig. 5.

cells in the string are fully discharged. The SOC deficit over pack lifetime is shown in Fig. 8a, b for the homogeneous and heterogeneous temperature packs, respectively. The SOC deficit is based on the per-cycle current throughput and fitted values to the reference capacity measurements at 0.1 C which provide an estimate of cell capacity per cycle (see Supplementary Section 2.2 for further details). Impedance growth in P20.0/0.0 (Fig. 8a) leads to a doubling of the SOC deficit from an average of 20% SOC in both cells to 40% SOC at EOL, accounting for the significant

difference in reference and dynamic capacity. Increasing the temperature of the homogeneous temperature packs non-linearly reduced the SOC deficit, reflecting non-linear changes in cell impedance with respect to temperature. SOC deficits estimated for the heterogeneous temperature packs (Fig. 8b) show similar trends, with low temperature and thus high impedance cells showing significant SOC deficits. In packs under applied thermal gradients, high temperature cells are discharged preferentially due to the decreased total branch resistance, and thus voltage cutoffs

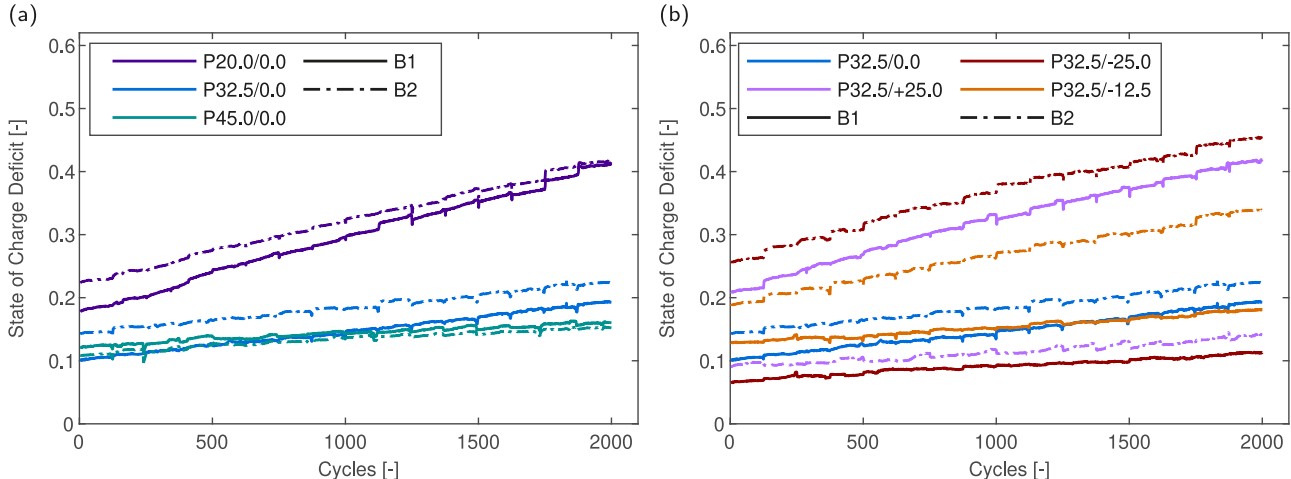

**Fig. 7 Schematic representation of the changes of current distribution during aging. a** Homogeneous temperature, start-of-life (SOL). **b** Homogeneous temperature, end-of-life (EOL). **c** Schematic representation of total branch resistance and branch mean current during homogeneous temperature aging. Dashed lines indicate start-, mid- and end-of-life, respectively. **d** Heterogeneous temperature, SOL. **e** Heterogeneous temperature, EOL. **f** Schematic representation of total branch resistance and branch mean current during heterogeneous temperature aging. Dashed lines indicate start-, mid- and end-of-life, respectively.

**Fig. 8 Estimated state-of-charge (SOC) deficit over lifetime for 1S2P packs. a** Homogeneous temperature packs. **b** Heterogeneous temperature packs.

are reached without the opportunity for SOC rebalancing. A 5% increase in the SOC deficit in the low temperature cells between P32.5/+25.0 and P32.5/-25.0 is also observed and is caused by the switching of interconnection resistances from assisting in balancing to causing an imbalance in total branch resistance as the direction of the applied thermal gradient switches. The magnitude of SOC deficit that a thermal gradient can cause is comparable to that caused by suboptimal temperatures and leads to a reduction in the accessible pack energy. Therefore, in order to maximise the energy available in a pack, minimising the thermal gradient is key, in addition to optimising the average cell temperature.

**Diagnosis of induced degradation modes**. To further analyse the cell-level degradation observed during the pack cycling, OCV analysis (using the methodology developed by ref. [27,28]—refer to Supplementary Section 2.3 for further details) was carried out using the 0.1 C reference cell discharge data obtained at each cycling breakpoint to identify the dominant degradation modes occurring. A RMSE error of <9 mV at SOL and <7 mV at EOL was achieved by the model in all cases, with errors likely due to the flat nature of the anode OCV curve, a general limitation of this diagnostic approach.

Figure 9a shows the identified losses caused by each of the three identifiable degradation modes at EOL for each cell cycled. In all cases, the dominant degradation modes are loss of lithium inventory (LLI) and loss of cathode active material ($LAM_{Ca}$). Loss of anode active material ($LAM_{An}$) in general contributes a significantly smaller amount to the overall cell degradation. A clear trend may be observed, with the dominant degradation mode in all cells cycled at low temperature (<32.5 °C) being $LAM_{Ca}$, whilst at higher temperatures (>32.5 °C) LLI becomes the dominant degradation mode. At intermediate temperatures (32.5 °C), approximately equal amounts of LLI and $LAM_{Ca}$ are observed. The degree of LLI is highly sensitive to temperature, as expected, ranging from 3.8% LLI in P32.5/+25.0 B1 to 13.5% LLI in P45.0/0.0 B1. Meanwhile increasing the cycling temperature leads to a more modest increase in loss of active material from both electrodes, with $LAM_{Ca}$ increasing from 6.2% in P32.5/+25.0 B1 to 10.1% in P45.0/0.0 B1. The range of identified $LAM_{An}$ was from 1.0% in P45.0/0.0 B1 to 3.9% in P32.5/−12.5 B1.

Figure 9 shows how cell degradation modes change with respect to temperature. Figure 9b shows the EOL SOH for each cell, which ranges from 96% to 86%. Here, the EOL SOH is well correlated with the cycling temperature, indicative of the dominance of temperature as a degradation stress factor. Considering the identified degradation modes (shown in Fig. 9c–e) with respect to temperature, as expected LLI is the dominant degradation mode at higher temperatures, and the amount of LLI identified is well correlated with temperature (Fig. 9c). Increasing temperature leads to increasing LLI, which is supported by the identified trends in $R_{SEI}$ growth and corresponds to accelerated SEI layer growth at elevated temperatures, although LLI may also occur via loss of lithiated active material. However, the OCV model is not capable of discrimination between loss of lithiated and delithiated active material.

Considering loss of active material in both electrodes, significantly more $LAM_{Ca}$ is observed than $LAM_{An}$ at all temperatures, with an insignificant correlation between the amount of $LAM_{Ca}$ and $LAM_{An}$ and temperature in all cases. However with increasing temperature the dominant degradation mode changes, with $LAM_{Ca}$ dominant for temperatures of <32.5 °C, and LLI dominant at temperatures of >32.5 °C. In all

cases the degree of $LAM_{An}$ is significantly lower, and the contribution of $LAM_{An}$ to overall cell degradation is limited. It is notable that in all packs cycled, the degree of $LAM_{Ca}$ identified is greater in the cell experiencing higher peak currents, and is evident in the 3 packs cycled under homogeneous temperature conditions, where in all cases B1 experiences both a higher peak current and increased $LAM_{Ca}$. The greatest difference in the homogeneous temperature packs is found in P45.0/0.0, in which B1 experienced significantly higher peak currents at start of discharge during cycling than B2.

Notably, the EOL SOH of the cycled cells is well correlated with the degree of LLI, due to the initial electrode active material loadings. The initial excess electrode capacity and cell stoichiometry means that loss of active material in both electrodes, of the order identified, does not lead to significant measurable loss of overall cell capacity. Note, this is only applicable when the amount of cell degradation is limited, as in this study where the maximum cell degradation is ~15%. However LLI, which leads to changes in cell stoichiometry and thus electrode alignment, leads to significant changes to the accessible cell capacity. Increasing electrode slippage, caused by LLI, leads to immediate reductions in cell capacity, as this slippage limits the capacity of each electrode available over the voltage range, and therefore LLI is the dominant cause of capacity loss during cycling.

**Discussion and proposed degradation mechanisms**. For packs cycled under homogeneous temperature conditions, both cell performance and the observed current distributions converged over the pack life. This can be attributed to the increasingly heterogeneous resistance growth of the cells, which effectively balances branch total resistances. In comparison, in the packs cycled under heterogeneous temperatures, divergence in cell operating conditions and cell performance was observed. Two principal factors cause this; the divergent cell impedance growth eliminating the rebalancing effects seen in the homogeneous temperature packs and increasingly differing cell capacities.

With applied thermal gradients, at SOL there is a mismatch between pack branch resistances, due to the decreased cell impedance at elevated temperatures. During aging, greater impedance growth is observed in the low temperature cells. Alongside slower cell impedance growth in the elevated temperature cells, these effects lead to increasingly unequal branch resistances during operation. Therefore during aging, the behaviour of the packs under thermal gradients becomes more heterogeneous, with greater SOC deficits and consequently reduced accessible capacity. These deficits in turn lead to higher mean SOCs at the lower cutoff voltage in low temperature cells and overall reduced depths of discharge, which would be a concern if cell lifetime is sensitive to the depth of discharge. Additionally, restricted depths of discharge within high SOC ranges may also accelerate degradation, due to the cell spending prolonged periods at high voltages. In the homogeneous cases, SOC deficits increased but convergently, due to increased cell impedances leading to increasingly rapid reaching of voltage limits. The second principal cause of divergence in the heterogeneous packs was the diverging rates of cell capacity loss, induced by temperature differences, causing mismatchs in cell voltages during discharge, consequently drive string currents through the low-resistance cells in the string.

Use of an OCV model identified, in all cases, limited loss of active anode material, with loss of active cathode and lithium inventory instead the dominant modes for all cells. In all cells, a balance of $LAM_{Ca}$ and LLI are observed. It is notable that for both $LAM_{Ca}$ and $LAM_{An}$ there is a limited correlation between the degree of each mode and temperature, indicating that the amount

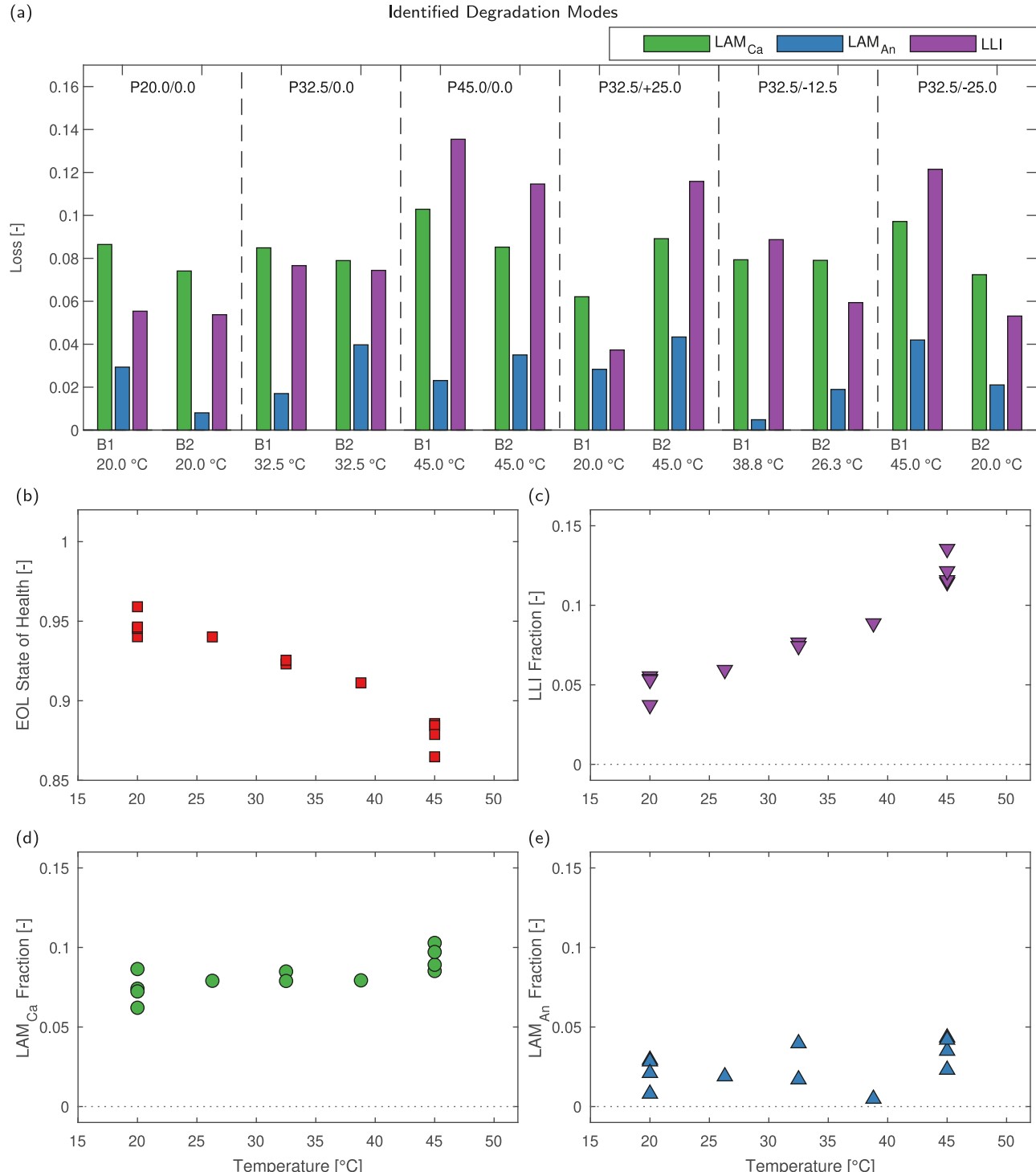

**Fig. 9 State of health (SOH) and identified degradation modes for all cells with respect to temperature after 2000 cycles. a** Fitted degradation modes identified using the open-circuit voltage (OCV) model for all cells. **b** State of health after 2000 cycles. **c** Fractional loss of lithium inventory after 2000 cycles. **d** Fractional loss of cathode active material after 2000 cycles. **e** Fractional loss of anode active material after 2000 cycles.

of each mode occurring is not significantly related to temperature over the range studied. However, this does not eliminate the possibility that the underlying mechanism leading to the degree of LAM changes with increasing or decreasing cell cycling temperature.

Considering the links between these mechanisms and degradation modes, ref. [32] provide a comprehensive review of the state-of-the-art understanding of the links between modes and

mechanisms for LIBs which can be applied to this study to link identified modes with possible degradation mechanisms.

Across all the packs studied, $LAM_{Ca}$ was found to be the dominant mechanism at low (20 °C) temperature, whilst LLI was the dominant mechanism at high (45 °C) temperature for all cells. The degree of $LAM_{Ca}$ was relatively insensitive to the cycling temperature (ranging from 8 to 11% $LAM_{Ca}$). In the low temperature aged cells, the dominant $LAM_{Ca}$ degradation mode is

likely caused by accelerated particle fracture and consequent pulverisation of cathode particles, leading to isolation of cathode particles. At the separator-cathode interface, local current densities are high due to through-plane electrode concentration gradients due to lower kinetics and reduced charge carrier mobility at lower temperature. Considering the rapid growth in cathode $R_{CT}$ at low temperature it is most likely that the key underlying degradation mechanism in the low temperature cycled cells is cathode particle fracture, with a degree of SEI layer formation leading to LLI (to a lower degree than in higher temperature cycled cells).

In comparison to the low temperature cells, the high temperature cells were shown to experience dominant LLI, with similar degrees of $LAM_{Ca}$ to the low temperature cells. SEI layer growth has been shown to be highly positively correlated with cycling temperature, as increased temperatures accelerate the unwanted side reactions which lead to formation of the SEI. Therefore, it is likely that the significantly increased amount of LLI identified in the high temperature cells is caused by SEI layer formation and growth, induced by the elevated temperatures. However, a similar amount of $LAM_{Ca}$ is identified in the high temperature cycled cells, but without an associated significant increase in cathode $R_{CT}$ identified using combined EIS and HPPC measurements. This is notable and indicative of a change in the underlying mechanism leading to the observed $LAM_{Ca}$ in the higher temperature cells compared those cycled at low temperature.

These competing mechanisms at low and high temperature are likely occurring simultaneously to differing degrees, with the dominant mechanism shifting as temperature increases. Post-mortem teardown images of cathodes cycled at elevated temperature (see Supplementary Fig. S17 clearly show evidence of particle fracture. However, by careful consideration of all the diagnostic evidence available (EIS, HPPC, OCV fitting and post-mortem imaging) it is possible to identify the most likely mechanisms underpinning degradation at all temperatures.

Overall, the above mechanisms are proposed to explain the observed behaviour of the cycled cells within the 6 packs tested. Generally, anode SEI layer growth has been considered to be the dominant mechanism for capacity loss and impedance growth. However, this study identifies cathode degradation as the dominant degradation mode present at intermediate temperatures (20.0–32.5 °C) which would generally be considered in a 'safe' operating temperature range. Whilst $LAM_{Ca}$ is measured at all temperatures studied, within the range of the study, significant changes to the fundamental cell capacity due to $LAM_{Ca}$ do not occur, due to the initial excess cathode present. If the study had continued to lower cell states of health, it would be expected that $LAM_{Ca}$ would lead to significant changes to fundamental cell capacity, once the initial excess cathode was consumed. However, $LAM_{Ca}$ leads to significant changes to cell kinetic performance, with significantly greater growth in cathode $R_{CT}$ at lower temperatures. This increasing cathode polarisation leads to a loss in accessible cell capacity and is typically not considered. As a consequence, the need to consider cathode degradation in detail when selecting pack designs and operating modes is highlighted by this work. It is noted that while the mechanisms previously proposed are supported based on EIS, HPPC and OCV diagnostic measurements taken during pack cycling, more detailed post-mortem analysis of cycled cells is required to conclusively demonstrate the underlying mechanisms leading to cathode damage.

Literature works, highlighted previously, have identified both divergent and convergent type behaviours in battery pack degradation. However, these prior studies have not provided robust underpinning understanding of the root cause of these effects, especially in the case of divergent degradation. In this study, we show that thermal gradients are a key driver for whether a battery pack operates in a convergent or divergent mode, with the often overlooked cathode degradation behaviour being a key driver. This was enabled by our experimental set-up which allowed for detailed mapping of the current disitributions over the life of the battery pack, as well as decoupled impedance measurements and degradation mode analysis.

However, we acknowledge that in real-world conditions the electrical, thermal and aging behaviour of a pack are coupled. In this work, we impose fixed thermal conditions in order to decouple this influence and to create highly controlled boundary conditions. Also, we suggest that a key overlooked influence in battery pack degradation is the evolution of cathode impedance growth, which we found in our NMC-LCO composite cathode cells. However, whether this generalises to other cathode chemistries remains an area of future work. Recommended areas of future study, therefore, include developing understanding of the feedback effects of evolving thermal boundary conditions induced by different aging paths, as well as exploring whether this effect generalises to other cathode chemistries.

## Conclusions
In this work, the effects of thermal gradients and temperature changes on the current distribution and degradation of parallel battery strings have been investigated. Detailed diagnostics were obtained at the single cell level by disassembling the parallel strings throughout their lifetime, allowing comprehensive examination of the degradation modes occurring within the cells during cycling. These identified modes are then associated with suggested underlying mechanisms and stress factors.

Application of thermal gradients to the packs led to loss of both instantaneous and lifetime performance compared to homogeneous temperature packs at the same mean temperature, highlighting the need to design for minimisation of thermal gradients alongside consideration of the interactions between pack topology and thermal boundary conditions. It was found that there were significant differences in static and dynamic capacity loss rates within homogeneous packs; with lower temperature cycled packs losing performance significantly more rapidly than those cycled at higher temperatures. This highlights the advantages of increased operating temperatures on instantaneous pack performance, due to the increased cell impedance at low temperature leading to increased SOC deficits at EOD. Conversely, the low temperature cycled packs exhibited minimal static capacity loss. Heterogeneous growth in the NMC-LCO cathode impedance at lower temperatures was shown to drive growing heterogeneities within parallel connected strings. Typically, optimal pack temperature is quoted in the range of 15–35 °C[33,34], however this work challenges the universal applicability of this assumption and highlights the need to capture cathode degradation processes; something which to-date is a gap in most modelling works.

It is proposed that the greater increases in cell impedance seen at low temperatures were caused by particle fracture and pulverisation due to high peak currents leading to an overall loss in electrochemically active cathode surface area. Meanwhile the loss of cathode active material at high temperature is more likely caused by structural disordering of the cathode, accelerated by high temperatures as opposed to mechanical damage. This proposed mechanism would explain the similar levels of cathode degradation observed at all temperatures, whilst also accounting for the different rates of growth of cathode charge transfer resistance observed at high and low temperatures.

In summary, this work presents a detailed analysis of parallel connected battery degradation under thermal gradients, which

has generally been lacking in the academic literature. Mechanistic models have been proposed which explain the increasingly divergent degradation behaviour under larger thermal gradients. Specific attention is placed on the role of the increasing cathode polarisation resistance, which was found to be inversely proportional to temperature, leading to a positive feedback effect. These results thus provide useful insights for battery pack designers; paving the way for more robust energy storage systems, as well as highlighting the need to accurate cathode degradation models. Future research directions include investigation of the effects of constant power and realistic drive cycle loading conditions on lifetime, and investigation of the effects of different cooling strategies and whether this observation generalises to other cathode chemistries.

## Data availability

All processed and unprocessed data generated as part of this work is available and can be found at https://doi.org/10.5281/zenodo.10207731. This includes raw time series data for all cell and pack cycling and reference performance tests (contained in Figs. 2, 3, 5, 6, 8, S1, S7) and all SOL/EOL impedance spectra (contained in Figs. 4 and S1) generated. All raw data is available in a .txt format and processed data is available in a .mat format.

## Code availability

All codes used in processing raw data and generating figures as part of this work is available and can be found at 10.5281/zenodo.10207731. The quantitative diagnostic codes used to identify degradation modes (shown in Fig. 9) is also available on reasonable request. Both data processing and diagnostic codes were generated and run using MATLAB r2020a.

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

## Acknowledgements

This work was kindly supported by the Innovate UK Battery Advanced for Future Transport Applications (BAFTA) project (104428), the EPSRC Faraday Institution's Multi-Scale Modelling Project (EP/S003053/1, grant number FIRG059) and the EPSRC Doctoral Training Partnership (EP/N509486/1 project number 1854850) for Max Naylor Marlow. The authors would also like to thank Gavin White, Oisin Shaw and Ryan Prosser for their generous assistance in developing the test bench which enabled this work, and Anisha Kanabar for her generous assistance in data presentation.

## Author contributions

M.N.M.: conceptualisation, data curation, formal analysis, investigation, methodology, visualisation, writing - original draft, writing - review & editing. J.C.: methodology, software, writing - review & editing. B.W.: conceptualisation, methodology, resources, supervision, funding acquisition, writing - review & editing.

## Competing interests

B.W. is a guest editor for the Communications Engineering Collection on Battery Management Systems for Vehicle Electrification, but was not involved in the editorial review of, nor the decision to publish, this article. Beyond this, the authors declare no known competing financial interests or personal relationships.
