## [Peer Review File · Communications Engineering]

Reviewers' comments:

Reviewer #1 (Remarks to the Author):

This paper investigates how the temperature variation impacts the degradation divergence among parallel-connected battery cells. This topic is important, and it is nice to see an experimental study, which is still rare in this field. The finding of the degradation mechanism in battery strings is useful. Overall, the paper is well written, but some issues need to be addressed, as listed below.

1. The logic of "impedance growth was negatively correlated with temperature, suggesting other dominant mechanisms" in the abstract is not clear to me before I look into the technical details.
2. The initial variation of the tested cells was strictly checked and tackled. I am just wondering whether the authors can provide some rough values about the initial variations among real-world industrial products (individual cells) from the perspectives of either capacity or internal resistance. Since the initial variation will exist anyway, it will be good to know the difference between the testing in this study and the real-world cases.
3. Even though some temperature differences were determined according to some cited experimental studies, the temperature difference among cells in the real battery pack highly depends on the cooling structures and the size of the parallel-connected battery strings. For example, Tesla Model 3 and Model S have different cooling structures, leading to different temperature variations among cells. I think the authors should at least justify that the chosen temperature differences can cover most real-world cases or even the worst case. In addition, one concern about the experimental study is that in real cases, the electrical-thermal-aging characteristics are concurrently intertwined, while in this experimental study, the thermal part is quite separate from the very beginning, thereby leading to a different phenomenon when compared to the variation progression in the real-world cases. For example, the temperature variation can increase first, stabilize later, and decrease over time in real battery packs due to the self-balancing mechanism among the parallel-connected battery cells, while this cannot be captured in the testing in this paper.
4. Since this study focuses on 2p1s configuration, I do not quite understand why both +25C and -25C are considered in the experiments, given that the initial variation among cells is well controlled. Shouldn't these two cases be the same? Is this because of the contact resistance distribution? Please explain.
5. How did the authors decide to check those individual cells every 125 cycles? I am wondering how the number 125 is figured out.
6. Regarding the EIS divergence at different frequencies, is there any electrode-level experimental support? It will be great to see some direct evidence of this interesting finding, as the current illustration is basically based on existing references.
7. Just my curiosity. Regarding the variation in the contact resistance, will it be much smaller in the commercial battery packs due to the more consistent assembly line? If so, what conclusions will be impacted or even changed if any?
8. Does the SoC deficit also consider the capacity measured at different temperatures to indicate how much percent of change is used in discharging? This is especially important when the SOC deficit in the heterogeneous temperature case is analyzed.
9. By the way, the SOC deficit is also influenced by the operation condition. For example, when the fully charging or fully discharging processes with a small cut-off current are involved in the testing, the SOC deficit can be regularly corrected. Is this true? Please elaborate on this, as the 2C cycles may not involve the CV process.

Reviewer #2 (Remarks to the Author):

This work presents an experimental study of thermal gradients on the lifetime degradation of parallel connected cells. Cycling experiments are conducted on 6 1S2P packs under homogeneous and heterogeneous temperatures range 20-45°C. The results of dynamic capacity and resistance changings are shown and discussed. By fitting cell OCV curves, the contributions of degradation mechanisms of LLI and LAM are quantitatively identified and reviewed for different operating thermal gradients. Overall, the work is interesting and is well written. It is valuable to Li-ion battery pack design. Some comments are as follows:

1. In Fig.2 c, there is P45.5/0.0 B1 that seems not the same as P45/0.0. What is the purpose and what can we see from the comparison between P45.5/0.0 B1 and P45.0/0.0 B2? Authors should give the explanation.
2. From Fig5 and Fig.6, we can see the DoD have significant effect on the current distributions between B1 and B2. What is the mechanism of it?
3. The degradation mathematic models should be given, thus we can understand how the data in Fig.9 are gotten quantitatively.
4. This study identifies cathode degradation as the dominant degradation mode present at intermediate temperatures. Whether this conclusion is still valid for LiFePO₄ or LiCoO₂ etc as cathode?

Reviewer #3 (Remarks to the Author):

In this paper, a mechanistic explanation of the observed degradation behaviour is proposed and the comprehensive aging dataset is made available. This work thus highlights the critical role of capturing cathode degradation processes in parallel-connected batteries, providing key insights for battery pack developers. I would like to put forward the following suggestions to improve the quality of the paper.

- (1) The novelty of the work should be explained in detail in the Abstract, Highlights, and Conclusion.
- (2) The expression of the abstract should be improved. A more detailed presentation of innovation should be conducted as well as the experimental verification.
- (3) More recent literature should be cited and analyzed, such as An improved feedforward-long short-term memory modeling method for the whole-life-cycle state of charge prediction of lithium-ion batteries considering current-voltage-temperature variation, Improved anti-noise adaptive long short-term memory neural network modeling for the robust remaining useful life prediction of lithium-ion batteries, and so on.
- (4) The expression logic can be improved for your own proposed method so that the innovation can be clearly understood. More expression of the mathematical analysis should be conducted to show your ideas clearly. Also, please try to highlight your proposed method and focus on it.
- (5) Grammar should be checked and improved for the whole content. Please try to make every sentence to be correct and easy to be understood.
- (6) More experimental result comparisons with references should be conducted for advantage discussion.
- (7) For this version, many contents are expressed for the traditional methods. Please pay more attention to your own content. Also, please express your innovation briefly for the Conclusion.

Response to reviewers

We would like to thank the reviewers for taking the time to assess this manuscript and for providing kind and helpful comments which have helped us refine and improve the manuscript. We have revised the manuscript in full and have provided 2 copies of the manuscript including a copy with changes highlighted for clarity and another version without highlighted edits.

The remarks from the reviewers and our response (blue text) (and specific highlighted changes) are detailed below, showing our specific response to each raised point which we hope will resolve these questions from the reviewers.

Response to Reviewer #1

This paper investigates how the temperature variation impacts the degradation divergence among parallel-connected battery cells. This topic is important, and it is nice to see an experimental study, which is still rare in this field. The finding of the degradation mechanism in battery strings is useful. Overall, the paper is well written, but some issues need to be addressed, as listed below.

Thank you for your kind comments - we have addressed each point raised individually as follows which we trust will resolve each point.

1. *The logic of “impedance growth was negatively correlated with temperature, suggesting other dominant mechanisms” in the abstract is not clear to me before I look into the technical details.*

Thank you for highlighting this. The abstract has been revised and we trust that this improves it's clarity.

2. *The initial variation of the tested cells was strictly checked and tackled. I am just wondering whether the authors can provide some rough values about the initial variations among real-world industrial products (individual cells) from the perspectives of either capacity or internal resistance. Since the initial variation will exist anyway, it will be good to know the difference between the testing in this study and the real-world cases.*

We have added a paragraph to the supplementary information (SI Page 1, Lines 11-18) which addresses this point. We note that the capacity variation observed in the cells used in this study were in line with typical values for commercial cells, the resistance variation was deliberately minimised to ensure that the trends in resistance growth were insofar as possible not a function of initial cell resistance, but induced by the current variation effects we investigated.

3. *Even though some temperature differences were determined according to some cited experimental studies, the temperature difference among cells in the real battery pack*

highly depends on the cooling structures and the size of the parallel-connected battery strings. For example, Tesla Model 3 and Model S have different cooling structures, leading to different temperature variations among cells. I think the authors should at least justify that the chosen temperature differences can cover most real-world cases or even the worst case. In addition, one concern about the experimental study is that in real cases, the electrical-thermal-aging characteristics are concurrently intertwined, while in this experimental study, the thermal part is quite separate from the very beginning, thereby leading to a different phenomenon when compared to the variation progression in the real-world cases. For example, the temperature variation can increase first, stabilize later, and decrease over time in real battery packs due to the self-balancing mechanism among the parallel-connected battery cells, while this cannot be captured in the testing in this paper.

Addressing the first point, we believe that the temperature range assessed covers a worst case scenario, and indeed in many applications smaller variation will be seen. However, considering the literature as a whole, it is clear that the magnitude of thermal heterogeneity will effect the magnitude of current variation, but not the overall effect.

We accept that by decoupling the thermal and electrical ageing of the cells this could impact the aging pathways present, however this paper is aimed at providing an exemplar and well controlled dataset/experiment which is why we chose to decouple thermal and electrical effects. Without decoupling these effects, it is very challenging to provide experimental data to investigate general trends in parallel string performance, as the performance of the coupled cooling system is naturally unique to that thermal management system design, which will impact results. We agree that this would be a useful area for further study and have added commentary on this as potential further work (Pages 19-20, Lines 509-526), but this coupling was deliberately excluded from this study.

- 4. Since this study focuses on 2p1s configuration, I do not quite understand why both +25C and -25C are considered in the experiments, given that the initial variation among cells is well controlled. Shouldn't these two cases be the same? Is this because of the contact resistance distribution? Please explain.*

Due to pack topology these cases are indeed different due to the interconnection resistances and position of the parallel string terminals. We refer the reviewer to E. Hosseinzadeh et al., Quantifying cell-to-cell variations of a parallel battery module for different pack configurations, Applied Energy, 2021, doi/10.1016/j.apenergy.2020.115859 for further illustration of the impact of topology selection.

- 5. How did the authors decide to check those individual cells every 125 cycles? I am wondering how the number 125 is figured out.*

The 125 cycle interval for reference measurements was selected to provide reasonably granular reference performance data on the individual cells, as it was not clear at the start of the study the likely duration of the cycling experiment. Selecting this frequency of checkups was also practically advantageous to managing the aging experiment and test channels, as it allowed a full round of aging and pack/cell reference data to be collected every 2 weeks.

- 6. Regarding the EIS divergence at different frequencies, is there any electrode-level experimental support? It will be great to see some direct evidence of this interesting finding, as the current illustration is basically based on existing references.*

Unfortunately we were not able to conduct any electrode level measurements of impedance for validation of these findings. However, we believe our approach of using a combination of EIS with HPPC measurements and carefully considering the evaluation time of the HPPC resistance, gives strong evidence of the divergence effect. The approach we used is based on the shown equivalence of single frequency EIS and HPPC evaluations from A. Barai et al., A study of the influence of measurement timescale on internal resistance characterisation methodologies for lithium-ion cells, Scientific Reports, 2018, doi/10.1038/s41598-017-18424-5.

7. *Just my curiosity. Regarding the variation in the contact resistance, will it be much smaller in the commercial battery packs due to the more consistent assembly line? If so, what conclusions will be impacted or even changed if any?*

Contact resistance is likely smaller in commercial packs where cells are joined using permanent techniques, but the interconnection resistance to the busbar (and therefore total branch resistance) may be higher depending on pack design. For example a typical wire bond in a cylindrical pack may have a more than a milliohm of resistance which is a large percentage of the cell resistance and will drive current distribution effects. We note that changes to the contact resistance will impact the magnitude of current distribution effects but will not whether the effect occurs. The review by Zwicker et al. Automotive battery pack manufacturing – a review of battery to tab joining. Journal of Advanced Joining Processes. 2020. <https://doi.org/10.1016/j.jajp.2020.100017>. gives a good overview of the contact resistances for various types of joining method.

8. *Does the SoC deficit also consider the capacity measured at different temperatures to indicate how much percent of change is used in discharging? This is especially important when the SOC deficit in the heterogeneous temperature case is analyzed.*

As we have available the slow rate capacity for the cells (measured on each individual cell at 0.1 C/20 degC at every reference breakpoint), the SOC deficit is calculated at each cycle by subtracting the actual charge passed from the theoretical capacity of the cells on each cycle. This is found by using a spline interpolation of the individual cell capacity loss data - as shown in Figure 2c/d the degradation observed is at no point significantly non-linear so this interpolation gives a good estimate of the static capacity of each cell during cycle aging and thus can be used to estimate the SOC deficit on each cycle. We agree that this was not clear in the manuscript and a comment (Page 13, Lines 354-356) has been added to make this clear along with a further explanation in the supplementary information (see SI Section 4)

On the point regarding the temperature at which the static capacity is measured, we agree that the temperature at which the static capacity is measured will have an impact on these estimates - however the cell used in the study is a very low impedance (<3 mOhm nominal) and the impact of the temperature difference on measured cell capacity judged to be small (measured at <1% for this cell, typically 4.88 Ah at 1 C/20 degC vs 4.92 Ah at 1 C/45 degC). It was not practically possible to measure the static capacity at all tested temperatures for all cells at each breakpoint in cycling, and it was judged that any inaccuracy in SOC deficit estimation caused by this effect is small compared to the magnitude of the SOC deficit

9. *By the way, the SOC deficit is also influenced by the operation condition. For example, when the fully charging or fully discharging processes with a small cut-off*

current are involved in the testing, the SOC deficit can be regularly corrected. Is this true? Please elaborate on this, as the 2C cycles may not involve the CV process.

In this study, cycling was done at constant current (no CV charging). We agree that the charging strategy will impact the amount of current rebalancing which is able to occur and may increase the magnitude of the current distribution somewhat. However, previous works have shown that a current distribution (and therefore SOC deficit) will be present no matter the charging strategy, due to the finite resistance of the string electrical connections. Furthermore, at every 125 cycles a set of characterisation tests were performed which allowed for the packs to rebalance.

Reviewer #2 (Remarks to the Author):

This work presents an experimental study of thermal gradients on the lifetime degradation of parallel connected cells. Cycling experiments are conducted on 6 1S2P packs under homogeneous and heterogeneous temperatures range 20-45°C. The results of dynamic capacity and resistance changings are shown and discussed. By fitting cell OCV curves, the contributions of degradation mechanisms of LLI and LAM are quantitatively identified and reviewed for different operating thermal gradients. Overall, the work is interesting and is well written. It is valuable to Li-ion battery pack design.

Thank you for your kind comments - we have addressed each point raised individually as follows which we trust will resolve each point.

1. *In Fig.2 c, there is P45.5/0.0 B1 that seems not the same as P45/0.0. What is the purpose and what can we see from the comparison between P45.5/0.0 B1 and P45.0/0.0 B2? Authors should give the explanation.*

Thank you for identifying this - "P45.5/0.0 B1" is an error and should refer to "P45.0/0.0 B1". This has been corrected in the revised manuscript

2. *From Fig5 and Fig.6, we can see the DoD have significant effect on the current distributions between B1 and B2. What is the mechanism of it?*

This is a commonly observed effect and is caused by the deviation in cell voltage and impedance as a function of cell SOC. It has been shown mathematically in other works (see B. Wu et al., Coupled thermal-electrochemical modelling of uneven heat generation in lithium-ion battery packs, Journal of Power Sources, 2013, doi/10.1016/j.jpowsour.2013.05.164) that if the interconnection resistance between cells is non-zero, some level of current heterogeneity will always occur, and therefore a difference in cell SOC will build, leading to a distribution of currents as the OCV and impedance of the different cells in the parallel string diverge. Rebalancing of currents occurs at low SOC where the non-linear increase in cell impedance at low SOC leads to a rapid increase in branch resistance and a subsequent rebalancing of the string current to branches with a lower total resistance

We have included a brief comment on prior work surrounding this effect (Page 11, Lines 277-279) in the manuscript

3. *The degradation mathematic models should be given, thus we can understand how the data in Fig.9 are gotten quantitatively.*

The diagnostic OCV-type model used in this work is not novel and thus is not the focus of the manuscript, but is based on work from Birkl et al. as follows:

- C. R. Birkl et al., A Parametric Open Circuit Voltage Model for Lithium Ion Batteries, Journal of The Electrochemical Society, 2015, A2271–A2280. doi:10.1149/2.0331512jes
- C. R. Birkl et al., Degradation diagnostics for lithium ion cells, Journal of Power Sources, 2017, doi:10.1016/j.jpowsour.2016.12.011)

The mathematics used to quantitatively assess degradation is presented in the supplementary information (see SI Section 5), including validation of our implementation of the model, and further underlying theory and validation of the approach can be found in the above references. We have amended the main manuscript to clearly direct to the SI for further details of the model used.

4. *This study identifies cathode degradation as the dominant degradation mode present at intermediate temperatures. Whether this conclusion is still valid for LiFePO₄ or LiCoO₂ etc as cathode?*

In this study a NMC-111/LCO blended cathode (approx 80/20 ratio) cell was selected for study. Without much more detailed analysis and experimental studies it is difficult to generalise this specific effect. This work however highlights the critical nature of considering cathode degradation at intermediate temperatures for pack designers. We have added this topic as a potential area for future work in the updated manuscript.

Reviewer #3 (Remarks to the Author):

In this paper, a mechanistic explanation of the observed degradation behaviour is proposed and the comprehensive aging dataset is made available. This work thus highlights the critical role of capturing cathode degradation processes in parallel-connected batteries, providing key insights for battery pack developers. I would like to put forward the following suggestions to improve the quality of the paper.

Thank you for your kind comments - we have addressed each point raised individually as follows which we trust will resolve each point.

1. *The novelty of the work should be explained in detail in the Abstract, Highlights, and Conclusion.*

Thank you for this comment. The abstract, highlights and conclusions have all been revisited to increase the clarity and highlight the novelty of the work.

2. *The expression of the abstract should be improved. A more detailed presentation of innovation should be conducted as well as the experimental verification.*

The abstract was rewritten to increase clarity and highlight the novelty of the work.

3. *More recent literature should be cited and analyzed, such as An improved feedforward-long short-term memory modeling method for the whole-life-cycle state of charge prediction of lithium-ion batteries considering current-voltage-temperature variation, Improved anti-noise adaptive long short-term memory neural network modeling for the robust remaining useful life prediction of lithium-ion batteries, and so on.*

Thank you for the suggestions. Our paper mostly focuses on understanding battery pack degradation behaviour and we have highlighted the most notable and recent works relevant to our studies which showcase both the divergent and convergent types of behaviour, whilst highlighting the gap in understanding of what causes this. Whilst the recommended papers are nice, we find that they focus more on the use of machine learning to forecast battery lifetime which is not the subject of our work and thus don't believe they are directly relevant here.

- 4. The expression logic can be improved for your own proposed method so that the innovation can be clearly understood. More expression of the mathematical analysis should be conducted to show your ideas clearly. Also, please try to highlight your proposed method and focus on it.*

Thank you for this comment. We have revised the entire manuscript to improve both the clarity of the communication but also highlight the novelty of the work. For the mathematical communication, please see our response to the other reviewers where we include additional clarification on our methodology.

- 5. Grammar should be checked and improved for the whole content. Please try to make every sentence to be correct and easy to be understood.*

Thank you for highlighting this. We have reviewed the entire manuscript and made changes throughout which are highlighted; improving the quality of the manuscript.

- 6. More experimental result comparisons with references should be conducted for advantage discussion.*

Thanks for this comment. In the literature review section we highlighted various studies which showcase both the divergent and convergent behaviours seen in battery packs but highlight the gap in understanding which was in part due to the inability to decouple the current distributions and impedances of the cells during aging, which we have done in this study. This has been added in the discussion section of the manuscript.

- 7. For this version, many contents are expressed for the traditional methods. Please pay more attention to your own content. Also, please express your innovation briefly for the Conclusion.*

Thank you. We have highlighted the novelty of our work which centred around the novel current measurement technique over the lifetime of the battery pack, in addition to the impedance measurements and degradation mode analysis which allowed for the identification of the often overlooked cathode degradation effects. Various other sections of the manuscript which include duplication of existing knowledge have been removed to streamline the narrative.

REVIEWERS' COMMENTS:

Reviewer #1 (Remarks to the Author):

The paper has been revised, and responses have been provided by the authors regarding those comments without any revision. So there is no further comment from me, and the paper can be accepted.

Reviewer #2 (Remarks to the Author):

I have no more comment.

Reviewer #3 (Remarks to the Author):

In this paper, the authors propose the root cause of the cathode impedance. The research content is well described for the revised version, including its innovation, logic, and experimental verification. It is suggested to be published.